# Climate Change Potentially Leads to Habitat Expansion and Increases the Invasion Risk of *Hydrocharis* (Hydrocharitaceae)

**DOI:** 10.3390/plants12244124

**Published:** 2023-12-10

**Authors:** Jiongming Yang, Zhihao Fu, Keyan Xiao, Hongjin Dong, Yadong Zhou, Qinghua Zhan

**Affiliations:** 1School of Life Sciences, Nanchang University, Nanchang 330031, China; yangjmncu@163.com (J.Y.); fuzhihao616@163.com (Z.F.); 2Hubei Xiuhu Botanical Garden, Xiaogan 432500, China; xiaoky@whu.edu.cn; 3Hubei Key Laboratory of Economic Forest Germplasm Improvement and Resources Comprehensive Utilization, Huanggang Normal University, Huanggang 438000, China; hongjind@hgnu.edu.cn

**Keywords:** species distribution models, aquatic plants, *Hydrocharis*, habitat expansion, invasion

## Abstract

Climate change is a crucial factor impacting the geographical distribution of plants and potentially increases the risk of invasion for certain species, especially for aquatic plants dispersed by water flow. Here, we combined six algorithms provided by the biomod2 platform to predict the changes in global climate-suitable areas for five species of *Hydrocharis* (Hydrocharitaceae) (*H. chevalieri*, *H. dubia*, *H. laevigata*, *H. morsus-ranae*, and *H. spongia*) under two current and future carbon emission scenarios. Our results show that *H. dubia*, *H. morsus-ranae*, and *H. laevigata* had a wide range of suitable areas and a high risk of invasion, while *H. chevalieri* and *H. spongia* had relatively narrow suitable areas. In the future climate scenario, the species of *Hydrocharis* may gain a wider habitat area, with Northern Hemisphere species showing a trend of migration to higher latitudes and the change in tropical species being more complex. The high-carbon-emission scenario led to greater changes in the habitat area of *Hydrocharis*. Therefore, we recommend strengthening the monitoring and reporting of high-risk species and taking effective measures to control the invasion of *Hydrocharis* species.

## 1. Introduction

Climate is a primary determinant influencing the geographical distribution of plants [1,2]. Previous studies have demonstrated that in the context of global warming, many plant species will shift their suitable habitats towards higher altitudes and latitudes [3,4]. Under future climate scenarios, certain plant species will experience an expansion in their distribution range, while others will undergo a reduction [5,6]. Changes in plant ranges have significant implications for the stability of ecosystems and the global pattern of biodiversity as well as pose new challenges for preventing species invasions [7].

With the changing climate, some dominant invasive plant species may gain increased opportunities for colonization and expansion [8,9]. These invasions pose a significant threat to biodiversity and ecosystem stability [10], thereby profoundly impacting human economies, health, and resource utilization [11,12]. Freshwater ecosystems are one of the main victims of climate change [13]. Research has demonstrated that aquatic plants have a greater likelihood of invading new environments when compared to their terrestrial counterparts due to their remarkable adaptability, rapid reproduction, and ability to disperse through water currents and other mechanisms [14]. Consequently, climate change may increase the risk of invasion by these species. Therefore, it is imperative to investigate the alterations in suitable habitats of aquatic plants under future climate change scenarios in order to effectively assess, predict, and manage biological invasions.

*Hydrocharis* (Hydrocharitaceae), a globally distributed aquatic plant genus with five accepted species, *H. chevalieri* (De Wild.) Dandy, *H. dubia* (Blume) Backer, *H. laevigata* (Humb. & Bonpl. ex Willd.) Byng & Christenh., *H. morsus-ranae* L., and *H. spongia* Bosc, has native ranges which span different continents without overlapping [15]. In the past, *H. laevigata* and *H. spongia* were treated as species in *Limnobium* Rich. Recent phylogenetic, morphological, and anatomical studies strongly support their close relationship with the three species of *Hydrocharis* L. [15,16,17]. Among *Hydrocharis*, *H. dubia*, *H. laevigata*, and *H. morsus-ranae* have been reported as alien species [18,19,20,21]. *H. chevalieri* is endemic to tropical Africa, and *H. spongia* is endemic to the Southeastern United States [22,23]. Efremov et al. (2020) studied the distributions of *H. chevalieri*, *H. dubia*, and *H. morsus-ranae* in the current climate, indicating that *H. dubia* and *H. morsus-ranae* have broad suitable areas and *H. chevalieri* has a narrow range worldwide [17]. However, there are still gaps in the knowledge regarding future dynamic changes in the habitat suitability for this genus under changing climatic conditions, which hinders effective invasion prevention measures.

Species distribution modeling (SDM) is extensively employed in the fields of ecology and biogeography [24], as it can be used to predict potential distribution areas and elucidate responses to climate change based on the potential relationship between species ecological niche and the environment. SDM effectively predicts changes in the distribution ranges of species under various climate change scenarios, thereby providing a scientific basis for conservation and invasion risk management [6].

Based on current distributed data of *Hydrocharis*, we conducted a comparative analysis of species distribution models in current and future climates for this genus to elucidate the response of *Hydrocharis* distribution ranges to climate change across different geographical regions worldwide. Specifically, we aimed to achieve the following objectives: (1) investigate the primary climatic factors that constrain the distribution of *Hydrocharis* species; (2) predict potential climate-suitable areas for each *Hydrocharis* species globally; and (3) explore dynamic changes in climate-appropriate areas of *Hydrocharis* under future climate scenarios.

## 2. Materials and Methods

### 2.1. Species Occurrence Records

We collected the distribution data of *Hydrocharis* from the Global Biodiversity Information Facility (GBIF, https://www.gbif.org/ (accessed on 26 July 2023)) [25], Integrated Digitized Biocollections (iDigBio, https://www.idigbio.org/ (accessed on 26 July 2023)) [26], and our own long-term field investigation in China from 2013 to 2019 [27]. In total, 64,683 occurrence records of *H. morsus-ranae*, 696 occurrence records of *H. laevigata*, 554 occurrence records of *H. spongia*, 379 occurrence records of *H. dubia*, and 48 occurrence records of *H. chevalieri* were obtained, respectively. We eliminated the error and non-native records based on the distribution area of these five species on the Plant of the World Online (POWO, https://powo.science.kew.org/ (accessed on 13 September 2023)). To avoid possible sampling bias in the occurrence records, we used the R packet “spThin” to filter the record points, set the minimum distance between the two sampling points to 10 km, repeated the calculation process 100 times, and finally chose the result of the 100th time [28]. Finally, 5584 points of *H. morsus-ranae*, 210 points of *H. laevigata*, 216 points of *H. dubia*, 269 points of *H. spongia*, and 34 points of *H. chevalieri* were used for modeling (Figure 1).

### 2.2. Environmental Variable Selection

We downloaded 19 bioclimatic factors based on the standard annual average of climatic conditions in the period 1970–2000 from WorldClim (http://www.worldclim.org (accessed on 26 July 2023)) as current climate data. For future climate scenarios, we used the results based on the CMCC Earth System (CMSC-ESM2) predictions, and we selected two shared socioeconomic pathways (SSPs), SSP1-2.6 and SSP5-8.5, which represent the highest- and lowest-carbon-emission scenarios [29]. Future climate data for the periods 2041–2060 (2050s) and 2061–2080 (2070s) were selected in this study. We also obtained altitude data from WorldClim, while all environmental variables were resampled to a spatial resolution of 5 arc-minutes (~10 km).

In order to avoid collinearity among variables, which may have resulted in overfitting of the model, we calculated the Spearman correlation coefficient (r) between each pair of variables. When the |r| between two variables was greater than 0.8, we removed the variable with a lower relative contribution rate. Finally, eight variables were retained for the prediction, i.e., mean diurnal range (Bio2), min temperature of the coldest month (Bio6), temperature annual range (Bio7), precipitation of the driest month (Bio14), precipitation seasonality (Bio15), precipitation of warmest quarter (Bio18), precipitation of the coldest quarter (Bio19), and elevation (Elev).

### 2.3. Construction and Evaluation of Species Distribution Model

We used six algorithms (CTA, FDA, GBM, GLM, MARS, and MAXENT) of the “biomod2” package to construct species distribution models. Biomod2 offers a comprehensive SDM tool that enables ensemble model prediction and result evaluation, overcoming the limitations associated with single models and enhancing the reliability of predictions [30]. For each species, 75% of the occurrence records were randomly selected as the training set, while the remaining 25% were used as the test set. Moreover, 1000 pseudo-absence points were randomly selected and repeated twice, and 5 rounds of cross-validation were carried out for each algorithm. Finally, 60 (6 × 2 × 5) base models were produced per species, and then we combined the models with high accuracy to obtain the optimal ensemble model (EMmodel). To assess the accuracy of the models, we used area under the curve (AUC) and true skill statistics (TSS) as evaluation indicators. AUC values ranged from 0.5 to 1, and TSS values ranged from −1 to 1; the closer the AUC and TSS values are to 1, the better the model performs [31,32].

### 2.4. Geographical Analyses

The suitability map obtained ranges from 0 to 1000,which indicated the species occurrence probability. We adopted a cutoff as the threshold, established a binary distribution map of presence and non-presence, and divided the suitability map into four levels, unsuitable (0~cutoff), low suitability (cutoff~600), moderate suitability (600~800), and high suitability (800~1000). In order to study the changes in suitable areas of *Hydrocharis* under future climate scenarios, we used SDMtools [33], a toolkit of ArcGIS, to compare the binary distribution maps under different climate scenarios and current climate scenarios, and statistically calculated the changes in suitable areas. To reveal the migration trend of each species’ climate-suitable ranges, we calculated the direction of center changes for each species by comparing the centroids of current and future binary distribution maps.

All analyses in this study were based on ArcMap 10.8 and R version 4.2.3.

## 3. Results

### 3.1. Model Performance and Predictor Variable Contributions

The results show that the EMmodel of all five species achieved very high AUC and TSS values, and the values of the EMmodel were higher than those of the individual algorithms (CTA, FDA, GBM, GLM, MARS, and MAXENT), demonstrating that the EMmodel obtained by combining different algorithms can predict results better (Appendix A, Table A1). The limiting factors for different species of *Hydrocharis* exhibited variations (Table 1). Temperature annual range (Bio7) was the most influential climate factor for *H. chevalieri* and *H. levigata*. Precipitation of warmest quarter (Bio18) and precipitation of the driest month (Bio14) were the climate factors with the greatest relative contribution to the distribution of *H. dubia* and *H. spongia*, respectively. The climate factor that played the largest role in the distribution of *H. morsus-ranae* was min temperature of the coldest month (Bio6).

### 3.2. Climate-Suitable Area of Five Hydrocharis Species

We categorized the predicted suitable zones into four levels: less than the cutoff for inappropriate zones, cutoff of 600 for general-suitability zones, 600 to 800 for moderate-suitability zones, and 800 or more for high-suitability zones (Figure 2). The main habitat of *H. chevalieri* is located in tropical Central Africa, primarily in Gabon, the Congo, and Uganda near the equator. However, there are also some climate-suitable areas on other continents at similar latitudes. The main habitat of *H. dubia* includes southeastern China, North and South Korea, Southeast Asia, and Japan. Climate-suitable areas of this species can also be found in Europe (Italy, Hungary, and Norway), northeastern Australia, and the Central and Southeastern United States. The main habitat of *H. laevigata* is scattered across South America and southern North America. Furthermore, there are additional climate-suitable areas for this species in Central America and Africa, along with Southeast Asia and coastal regions of Australia. The main habitat of *H. morsus-ranae* encompasses most parts of Central and Western Europe, as well as western Russia, and it also has a secondary habitat in North America. The main habitat of *H. spongia* is concentrated in the Southeastern United States, with extensive areas exhibiting high climate suitability in Argentina and Uruguay. Meanwhile, a few pockets of climate suitability for this species exist in Western Europe.

### 3.3. Change in Climate-Suitable Area of Hydrocharis in Future Climate Scenarios

By comparing the predicted outcomes across various future climate scenarios and the current climate scenario, we obtained the patterns of change in suitable climatic areas for the five *Hydrocharis* species (Figure 3). Simultaneously, we calculated alterations in both expanded and reduced areas (Figure 4).

Specifically, the climate-suitable areas of *H. dubia*, *H. morsus-ranae*, and *H. spongia*, whose three main distribution regions are located in the Northern Hemisphere, showed a significant expansion trend, and the expansion area was mainly located in the higher latitude of the Northern Hemisphere. In contrast, the climate-suitable area of *H. chevalieri* did not change much and was slightly reduced. It is worth noting that the non-native suitable area of *H. chevalieri* showed a significant contraction trend, but its native distribution in Central Africa only partially decreased in the West African region, such as in Libya, Cote d‘Ivoire, and Ghana, and significantly expanded in the southern part of the native suitable area. The area of *H. laevigata* experienced a similar contraction and expansion, and the overall distribution area will remain stable in the future climate scenario, where the contraction area is mainly located in the suitable areas of the Northern and Southern Hemispheres, while the expansion area is mainly located in the lower latitudes of the tropics and subtropics.

In summary, the change in climate-suitable areas of *Hydrocharis* species was more pronounced under the high-carbon-emission scenario (SSP5-8.5) compared to the low-carbon-emission scenario (SSP1-2.6), with a greater magnitude observed in 2070 as opposed to 2050.

### 3.4. Center Shift of Climate-Suitable Area of Hydrocharis

By analyzing the change in the centroid in the future climate-suitable area, we can reveal the migration trend of the *Hydrocharis* genus with climate change (Figure 5). For *H. dubia*, *H. morsus-ranae*, and *H. spongia*, which are mainly distributed in the Northern Hemisphere, their centroids have a tendency to move to the northwest in a high-latitude direction, where *H. dubia* has the largest migration distance, while *H. spongia* has the least obvious migration trend to a higher latitude. As for the remaining two species, which are mainly distributed in low latitudes, their latitudes do not change much under future climate change scenarios. *H. chevalieri* migrates to different longitudes under different carbon emission scenarios, and in the case of SSP5-8.5, its distribution area has a small trend of southward migration by 2070. For *H. laevigata*, the centroid of its climate-suitable area has an obvious trend of westward movement in the future.

## 4. Discussion

*Hydrocharis* is a widely distributed aquatic genus observed across the world and includes some species with a narrow distribution and some widespread ones with potential invasion risks; thus, the impact of future climate change on this genus is worth exploring. In this study, we predicted the climate-suitable area changes in all five species of *Hydrocharis* under current and future carbon emission scenarios. Our results show that *H. dubia*, *H. morsus-ranae*, and *H. laevigata* had a wide range of suitable areas in the world and were prone to invasion risk, while the suitable areas of *H. covalieri* and *H. spongia* were more localized. Additionally, in order to adapt to future climate change, the five species of *Hydrocharis* exhibited different responses, but overall, *Hydrocharis* was predicted to experience an expansion of its climate-suitable areas.

### 4.1. Climate-Suitable Area of Hydrocharis on a Global Scale

Our study expanded the modeling for all five accepted *Hydrocharis* species. For the three overlapping species, our results are largely consistent with a previous study, showing a narrow potential range for *H. chevalieri* restricted to Central Africa, and broad climate-suitable areas globally for the widespread species *H. dubia* and *H. morsus-ranae* (Figure 2) [17]. The consistency across independent modeling efforts using different methods increases confidence in the predicted distribution patterns. These results together confirm that *H. dubia* and *H. morsus-ranae* have strong adaptability and diffusion ability and have a high risk of becoming invasive globally [17].

For the other two previously unmodeled species, we found that in addition to a limited range mainly in the Southeastern United States, where *H. spongia* originated, there are also some suitable areas of high suitability in South America (Figure 2), but no records of *H. spongia* have been found in this region as of yet (Figure 1). This suggests that dispersal limitation is an important factor in shaping the geographical pattern of this plant [34,35]. *H. laevigata* showed a very broad potential distribution spanning much of the Americas as well as parts of Africa, Southeast Asia, and Australia, which is consistent with the documented invasion risk of *H. levigata* in the past [36,37], reflecting its ability to naturalize widely beyond its native subtropical–tropical American range [38]. 

In summary, modeling these additional species expanded our knowledge of *Hydrocharis*’s distribution and invasion risk. Our global models for *Hydrocharis* species provide useful information to guide monitoring of range shifts and introduced populations. 

### 4.2. Dynamic Change in Potential Suitable Area of Hydrocharis in Future Climate

The models we employed projected predominantly northward shifts for northern temperate *Hydrocharis* species (*H. dubia*, *H. morsus-ranae*, and *H. spongia*) in the future (Figure 3 and Figure 5), following the warmer conditions expected under climate change [4]. Among them, *H. dubia* had the most obvious climate-suitable area dynamics, gaining many new suitable zones in Russia, Europe, and North America. In contrast, *H. morsus-ranae* and *H. spongia* were more inclined to expand at the edges of their original suitable area. Their northward migration was not as dramatic as that of *H. dubia*, but warming still made more areas suitable for them. Previous studies have shown that *H. dubia* prefers relatively warmer temperatures compared to *H. morsus-ranae* [17], which may limit the distribution of *H. dubia* in regions with high latitudes. However, with gradual climate warming and increased precipitation, these high-latitude regions are becoming suitable for the survival of *H. dubia*. In comparison, the slower habitat expansion of *H. spongia* may reflect its narrower temperature and precipitation tolerance.

The two species mainly distributed in tropical and subtropical regions showed different distribution dynamics (Figure 3 and Figure 5), and they showed no obvious trend of moving to higher latitudes. This may be related to the complexity of climate change in tropical regions. Specifically, future warming in tropical and subtropical regions is likely to differ from that in northern regions, with regional variations in rainfall [39]. In addition, tropical species are more sensitive to factors such as sunlight and climate seasonality [40]. Combined with a variety of uncertainties, habitat changes in low latitudes may show a more complex pattern not only related to temperature or precipitation changes [41]. Future studies need to consider the combined effects of multiple climate variables in the tropical region and reduce the uncertainty of simulations in order to more accurately predict the response of and distribution changes in tropical species.

In general, under the scenario of high carbon emissions, the dynamic change in the suitable area of *Hydrocharis* will be more drastic, and this effect will be more obvious over time. Similar results have been found in some studies on changes in the future climate-suitable areas for plants [42,43]. This phenomenon may be attributed to the amplified climatic fluctuations observed in the high-carbon-emission scenario [39], which subsequently elicit a more pronounced positive response from vegetation [44]. These findings suggest that future carbon emissions resulting from human social activities will have a profound impact on the spatial distribution of certain plant species and elevate their susceptibility to invasion.

### 4.3. Invasion Risk and Management of Hydrocharis Species

Not all climate-suitable areas of *Hydrocharis* have been introduced, but with the increasing frequency of human social activities, the introduction of alien species caused by humans has become more frequent [12]. For example, *H. laevigata* is often introduced as an ornamental plant in aquariums and gardens. Thanks to its extensive ecological adaptability and strong reproductive capacity [45], when it escapes into the wild, it can quickly settle in new habitats [38] and deteriorate the water quality of the habitats. As a result, the habitat is no longer suitable for other aquatic plants and animals [19]. At the same time, past studies have shown that bird migration and seed migration with water currents may also be important causes of aquatic plant invasion [46]. A similar phenomenon has been observed in *H. morsus-ranae*, which has invaded parts of North America, such as the Rideau and Ottawa River systems, the St. Lawrence River, Lake Ontario, Lake Erie, Lake Kawasa, and other river and lake systems in Canada [21,47,48], Our projections also confirm that the climate-appropriate area in this region will expand in the future. Once *H. morsus-ranae* invades a new aquatic ecosystem, it has the potential to establish a dense mat of interwoven leaves and roots, impeding light penetration and hindering water flow. This can lead to reduced nutrient and oxygen availability for native species, resulting in decreased dissolved oxygen (DO) concentrations. Furthermore, it can cause detrimental effects on the diversity of indigenous fish and plants, as well as disrupt overall ecosystem stability [49,50]. Therefore, it is of great significance to monitor and prevent the invasion in climate-suitable areas.

We assessed the risk of invasion of *Hydrocharis* on a global scale and the species dynamics under future climate scenarios. Based on the predicted results, we believe that *H. dubia*, *H. morsus-ranae*, and *H. laevigata* have a high invasion risk, and this risk is likely to increase with the intensification of climate warming (especially for *H. dubia* and *H. morsus-ranae*). These species thus need to be paid attention to and managed. We recommend strengthening the monitoring and reporting of these species in non-native areas to detect and control their spread in a timely manner. At the same time, in some high-risk areas, trade and transport of these species can be restricted or banned if necessary. Finally, we recommend effective control measures to eliminate or reduce the impact of these species on invasive areas [47]. In contrast, *H. chevalieri* and *H. spongia* species have a lower risk of invasion because their climate-suitable areas are narrow and confined to specific geographical areas. However, this does not mean that their potential threat to non-native areas can be ignored, as climate change may alter the extent and location of their habitat areas, and they may also have adverse environmental effects when they enter new areas.

## 5. Conclusions

Using the EMmodel provided by the biomod2 platform, this study predicted changes in climate-suitable areas and the native distribution of five *Hydrocharis* species under the current climate and two future carbon emission scenarios and assessed their invasion risk on a global scale. Our results show that *H. dubia*, *H. morsus-ranae*, and *H. laevigata* in the genus *Hydrocharis* have a wide range of suitable habitat areas in the world and are prone to invasion risk. The habitats of *H. chevalieri* and *H. spongia* are narrow and limited to specific geographical areas. At the same time, we found that the five species of *Hydrocharis* exhibited different responses to future climate changes, but overall, *Hydrocharis* will have a wider range of suitable areas. In addition, in the high-carbon-emission scenario, the suitable areas of the *Hydrocharis* species showed greater changes than those in the low-carbon-emission scenario. Based on these results, we have made a number of targeted management recommendations, including strengthening monitoring and reporting, restricting or banning trade and transportation, and adopting effective control measures. We hope that this study can provide a reference and inspiration for the conservation and management of *Hydrocharis* species and other aquatic plant species in the future.

## Figures and Tables

**Figure 1 plants-12-04124-f001:**
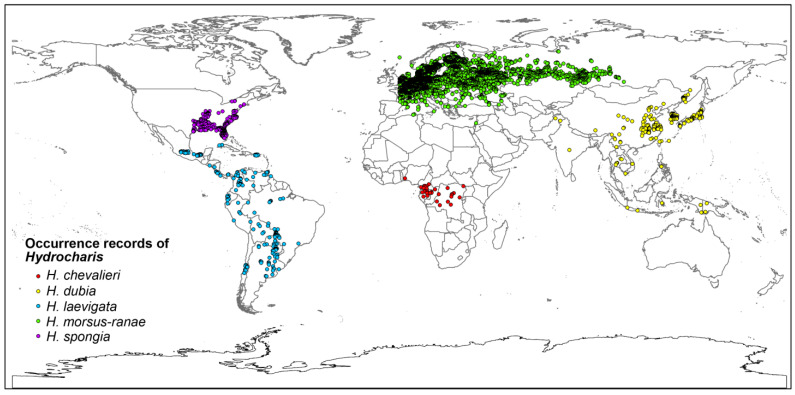
Occurrence records of *Hydrocharis* used for SDM, excluding non-native records and filtered by “Spthin”.

**Figure 2 plants-12-04124-f002:**
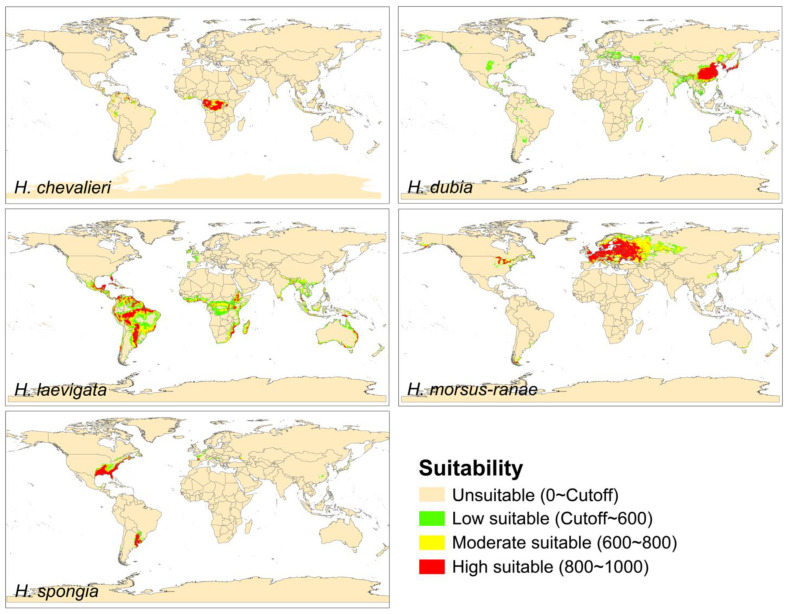
Predicted climate-suitable area of *Hydrocharis* in the current period.

**Figure 3 plants-12-04124-f003:**
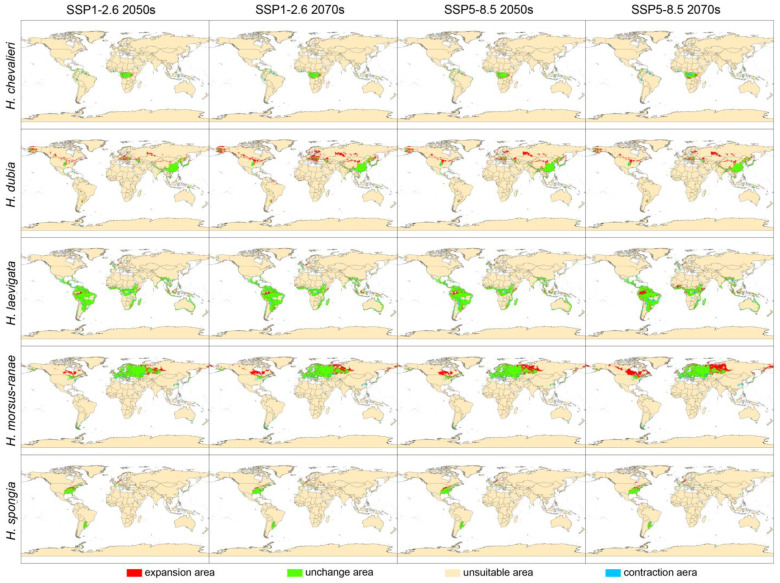
Dynamic changes in climate-suitable area of *Hydrocharis* under future climate scenarios.

**Figure 4 plants-12-04124-f004:**
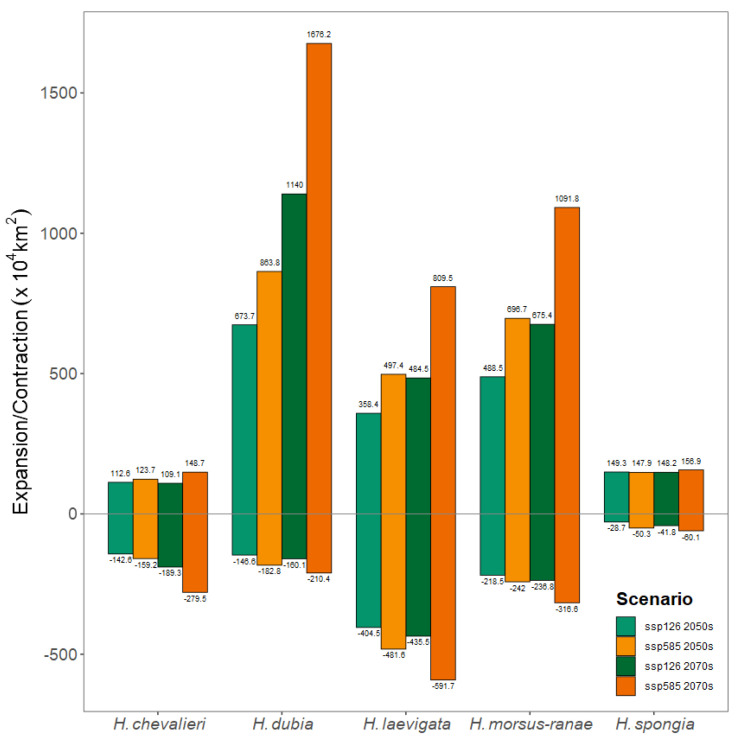
Expansion of and contraction of climate-suitable area of *Hydrocharis*.

**Figure 5 plants-12-04124-f005:**
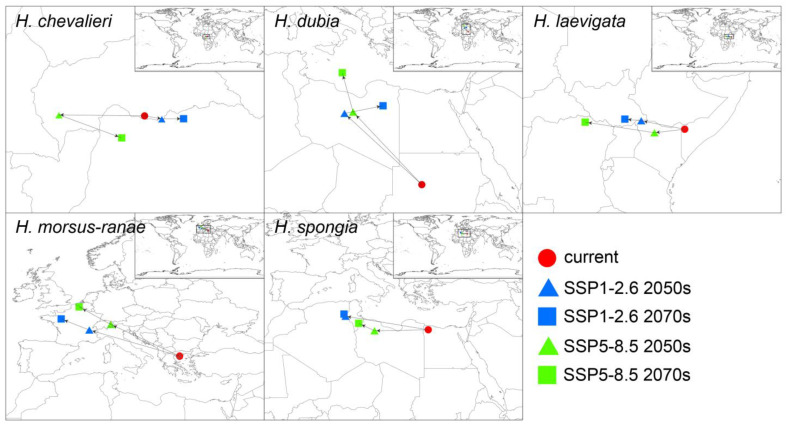
Center changes in climate-suitable area for *Hydrocharis*.

**Table 1 plants-12-04124-t001:** Relative contributions of selected predictor variables in the ensemble model.

	*Hydrocharis chevalieri*	*H. dubia*	*H. laevigata*	*H. morsus-ranae*	*H. spongia*
Bio2	1.60%	7.45%	11.88%	14.11%	12.87%
Bio6	12.78%	17.32%	13.86%	49.15%	2.46%
Bio7	55.26%	14.16%	42.44%	1.78%	12.66%
Bio14	0.59%	0.54%	7.57%	0.86%	41.60%
Bio15	0.93%	3.37%	1.12%	6.36%	1.31%
Bio18	1.30%	54.07%	9.47%	9.00%	7.00%
Bio19	1.54%	1.21%	0.76%	1.78%	1.27%
Elev	26.00%	1.87%	12.90%	16.96%	20.82%

## Data Availability

All generated data are included in this article.

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
