# Peer review of "Climate Change Potentially Leads to Habitat Expansion and Increases the Invasion Risk of Hydrocharis (Hydrocharitaceae)"

_plants, 2023, doi:10.3390/plants12244124_

Round 1

Reviewer 1 Report

Comments and Suggestions for Authors

Line 50. Include for the first time the authors of Hydrocharis species.

Line 51. Citing [16] for Hydrocaris geographical distribution, it would be better to add [24], as some of the species treated here are missing in both works.

Line 80. Regarding to Lines 84-85 (down), perhaps the arguments should be indicated in the application of the R packet "spThin".

Lines 84-85. The distribution map in Fig. 1 is the result of applying the R package "spThin" on the GBIF data of native distribution of Hydrocharis. This should be informed to avoid the reader thinking that this figure shows the complete native range of the species (as an example, native populations of H. morsus-ranae -200 GBIF records at https://doi.org/10.15468/dl.mnh45d- from Spain, Portugal and Morocco- have been removed from the map and the analysis).

Line 202 (for discussion section). It has not been sufficiently discussed how the centroids of two species (H. dubia and H. spongia), at present and for the different scenarios chosen, have undergone a significant shift towards North Africa from their native regions (E Asia and E Europe).

Line 208. Correct "H. Laviegata" to "H. laviegata", in italic...

Lines 261-262. The conclusion "which subsequently provoke a more pronounced positive response from vegetation" should be supported by a reference.

Reviewer 2 Report

Comments and Suggestions for Authors

The authors present a very interesting article focusing on the increased risk of invasion for certain aquatic plants under climate change scenarios. The methodology is robust, presented in great detail and have been used in may other similar studies, mostly on terrestrial plants.

In my opinion the presented work  is  important and very useful for many other scientists that intent to work on SDMs and aquatic plants. Therefore, I recommend to consider this article for publication following a minor revision.

I have only a few comments.

The abstract could be a bit more informative, particularly about the methods. I think it is worth mentioning the methodology for using SDMs and the ensemble modelling.

Line 43: Shouldn’t be SDMs are not is?

L44: they integrate?

L43-47: I suggest to move the small paragraph about SDMs after the paragraph about the Hydrocharis genus

L127: Overall the methodology is quite detailed and very nicely presented. The packages and the methods described for the SDMs are well known and widely used in similar studies. However at this point  I did not fully understand what you mean by «the center of current». I would appreciate if  you could rewrite more accurately how you revealed the migration trend for each species.

L135: I think there is an issue of expression here. Better wording is recommended.

The presentation of the results is very nice. The maps and the figures are of high quality and easy to interpret.

Concerning the discussion I think that the authors could add some information about the invasiveness of these species. Since they focus on the invasion risk it would benefit the article if there was some discussion about known cases where these species invaded ecosystems and why or how. It would be great if the article showed that these plants are indeed potential invaders by showcasing examples from the literature. Further discussion could concern the trend of expansion by comparing known cases with the modelled outcome.

Comments on the Quality of English Language

The manuscript is very well written. I detected minor language issues so I recommend a careful linguistic revision by the authors to correct any small errors.  The structure is ok and can be easily followed by the reader
